# Mammalian forelimb evolution is driven by uneven proximal-to-distal morphological diversity

**Priscila S Rothier[1]\*, Anne-Claire Fabre[2,3,4], Julien Clavel[4,5], Roger BJ Benson[6], Anthony Herrel[1]**

[1]Département Adaptations du Vivant, Muséum National d'Histoire Naturelle, Paris, France; [2]Naturhistorisches Museum Bern, Bern, Switzerland; [3]Institute of Ecology and Evolution, University of Bern, Bern, Switzerland; [4]Life Sciences Department, Vertebrates Division, Natural History Museum, London, United Kingdom; [5]Université Claude Bernard Lyon 1, CNRS, ENTPE, UMR 5023, Villeurbanne, France; [6]Department of Earth Sciences, University of Oxford, Oxford, United Kingdom

**Abstract** Vertebrate limb morphology often reflects the environment due to variation in locomotor requirements. However, proximal and distal limb segments may evolve differently from one another, reflecting an anatomical gradient of functional specialization that has been suggested to be impacted by the timing of development. Here, we explore whether the temporal sequence of bone condensation predicts variation in the capacity of evolution to generate morphological diversity in proximal and distal forelimb segments across more than 600 species of mammals. Distal elements not only exhibit greater shape diversity, but also show stronger within-element integration and, on average, faster evolutionary responses than intermediate and upper limb segments. Results are consistent with the hypothesis that late developing distal bones display greater morphological variation than more proximal limb elements. However, the higher integration observed within the autopod deviates from such developmental predictions, suggesting that functional specialization plays an important role in driving within-element covariation. Proximal and distal limb segments also show different macroevolutionary patterns, albeit not showing a perfect proximo-distal gradient. The high disparity of the mammalian autopod, reported here, is consistent with the higher potential of development to generate variation in more distal limb structures, as well as functional specialization of the distal elements.

**\*For correspondence:** priscilasrd@gmail.com

**Competing interest:** The authors declare that no competing interests exist.

## Editor's evaluation

This study reports an interesting analysis of evolutionary variation in forelimb/hand bone shapes in relation to functional and developmental variation along the proximo-distal axis. The authors found expected and compelling patterns of evolutionary shape variation along the proximo-distal axis but less expected, yet equally compelling, patterns of shape integration. This paper will be of interest to researchers working on macroevolutionary patterns and sources of morphological diversity.

## Introduction

The evolutionary origin of limbs sets the stage for the remarkable ecological diversity of Tetrapoda (*Shubin et al., 1997*). From delicate wings to powerful excavating claws, from slender hooved legs to wide flattened flippers, limb formation is intrinsically integrated with and constrained by the determination of the tetrapod body plan (*Raff, 1996*). The tetrapod limb is typically composed of three

**Figure 1.** Forelimb diversity of mammals. The topology includes all genera examined in this work, representing the exceptional forelimb morphological variation for some of the species analysed. The topology was estimated using maximum clade credibility from a posterior sample of 10,000 trees published by *Upham et al., 2019*.

basic components: the proximal stylopod (upper arm and thigh), the intermediate zeugopod (lower arm and calf), and the distal autopod (hand and foot). The proximal to distal organization of segments is correlated with their respective evolutionary appearance, the stylopod being the first structure to evolve, later followed by the zeugopod, and finally the autopod (*Shubin et al., 1997*). Although the three-segment pattern is conserved among quadruped tetrapods, the morphology of these structures along the proximo-distal axis may evolve differently among groups (*Cooper et al., 2011*; *Galis et al., 2001*; *Holder, 1983*; *Sears et al., 2007*).

Limbs are often studied for their exceptional morphological and ecological diversity (*Chen and Wilson, 2015*; *Grizante et al., 2010*; *Kohlsdorf et al., 2001*; *Ledbetter and Bonett, 2019*; *Polly, 2007*; *Rothier et al., 2022*; *Rothier et al., 2017*; *Stepanova and Womack, 2020*). In mammals, for

example, the forelimb is present in all species and is typically more variable than the hind limb, possibly due to its greater number of functional roles (*Figure 1*; *Polly, 2007*; *Schmidt and Fischer, 2009*). The meristic composition of tetrapod forelimb segments varies along the proximo-distal limb axis, where the autopod exhibits most of the diversity in terms of the number and position of skeletal elements (i.e. fusion and loss of carpal and tarsal bones and alteration of the phalangeal formula; *Cooper et al., 2007*; *Hamrick, 2001*; *Holder, 1983*; *Luo et al., 2015*; *Saxena et al., 2017*). Except for lineages that have undergone complete limb loss such as snakes and caecilians, the meristic composition of proximal segments is much more conserved than that of the autopod, displaying some but less frequent cases of element reduction and partial fusion of the zeugopod bones (observed in anurans, bats, manatees, horses, etc., *Holder, 1983*; *Keeffe and Blackburn, 2022*; *Sears et al., 2007*). Although this meristic information is useful to quantify major evolutionary changes in element composition, most of the morphological variation observed in the limbs results from changes in the shape and relative size of individual elements (i.e. variation of form) without changing the numbers of elements, and is often associated with functional adaptation (*Fabre et al., 2013*; *Fabre et al., 2015*; *Janis and Martín-Serra, 2020*; *Lungmus and Angielczyk, 2021*; *Maier et al., 2017*; *Sears et al., 2018*). Despite its importance, it remains unclear how this macroevolutionary variation of form is partitioned between the three limb segments.

Both functional and developmental factors predict that distal elements should show greater variation of form than more proximal elements. Developmental mechanisms predict this pattern due to the timing and spatial structure of morphogenesis, which has been suggested to influence the macroevolutionary outcome of adult morphologies, including that of the skull (*Bardua et al., 2021*; *Fabre et al., 2020*), the vertebrae (*Adler et al., 2022*), and the limbs (*Holder, 1983*; *Stepanova and Womack, 2020*). Each limb initiates as a bud that extends from the body wall and where skeletal elements are generally specified in a proximal to distal sequence that matches their evolutionary appearance during tetrapod origins: development begins with the stylopod, followed by the zeugopod, and terminating in the autopod at the distal end (*Figure 2A*; *Schneider and Shubin, 2013*; *Shubin et al., 1997*; *Stopper and Wagner, 2005*). Limb development has been notably studied in mammals, revealing that different species have more similar forelimb morphology during early development, and become more disparate during later stages of morphogenesis (*Ross et al., 2013*). Likewise, gene expression of different mammal species is more conserved during early phases of limb development, compared to later phases (*Maier et al., 2017*), and these patterns might reflect the intrinsic temporal properties of embryogenesis (*Galis et al., 2001*; *Sears et al., 2018*).

The timing of development has been already suggested to impact the uneven diversity and evolution of limb segments in frogs, with distal, late-developing bones being more variable and tending to diversify faster than proximal, early-forming elements (*Stepanova and Womack, 2020*). Indeed, early developmental processes mediating the initial specification of structures are generally more constrained than those governing later events, such as organ specialization (*Kalinka and Tomancak, 2012*). Therefore, because limb development proceeds proximo-to-distally, developmental perturbations at later phases may tend to accumulate higher morphological variation in distal elements (*Hallgrímsson et al., 2002*). One way to investigate the levels of developmental and functional constraints on adult morphologies is by quantifying the phenotypic integration among traits, inferred from the covariation between structures. For example, because the fore and hind limbs are serially homologous, they share genetic and developmental processes that give rise to strong phenotypic integration between and within the limbs (*Ruvinsky and Gibson-Brown, 2000*; *Young and Hallgrímsson, 2005*). In mammals, the correlation between homologous limb segments of the fore- and hind limbs (i.e. humerus with femur, radius with tibia, metacarpal with metatarsal) suggests that proximal segments are highly integrated to each other (*Hallgrímsson et al., 2002*; *Schmidt and Fischer, 2009*; *Young and Hallgrímsson, 2005*). In contrast, the more distal elements of the hand and foot show more variable patterns of integration, which may reflect functional specialization and the accumulation of variation during later phases of development (*Hallgrímsson et al., 2002*; *Rolian, 2009*; *Young and Hallgrímsson, 2005*). A consequence for limb diversification is that the patterns and pace of morphological evolution might not be the same between proximal and distal segments.

Here, we investigate the evolutionary patterns underlying the morphological diversification of mammalian forelimb segments along a proximal-to-distal axis, using a comprehensive data set of 638 species, capturing over 85% of Mammalia family-level diversity (*Supplementary file 1*). We ask to

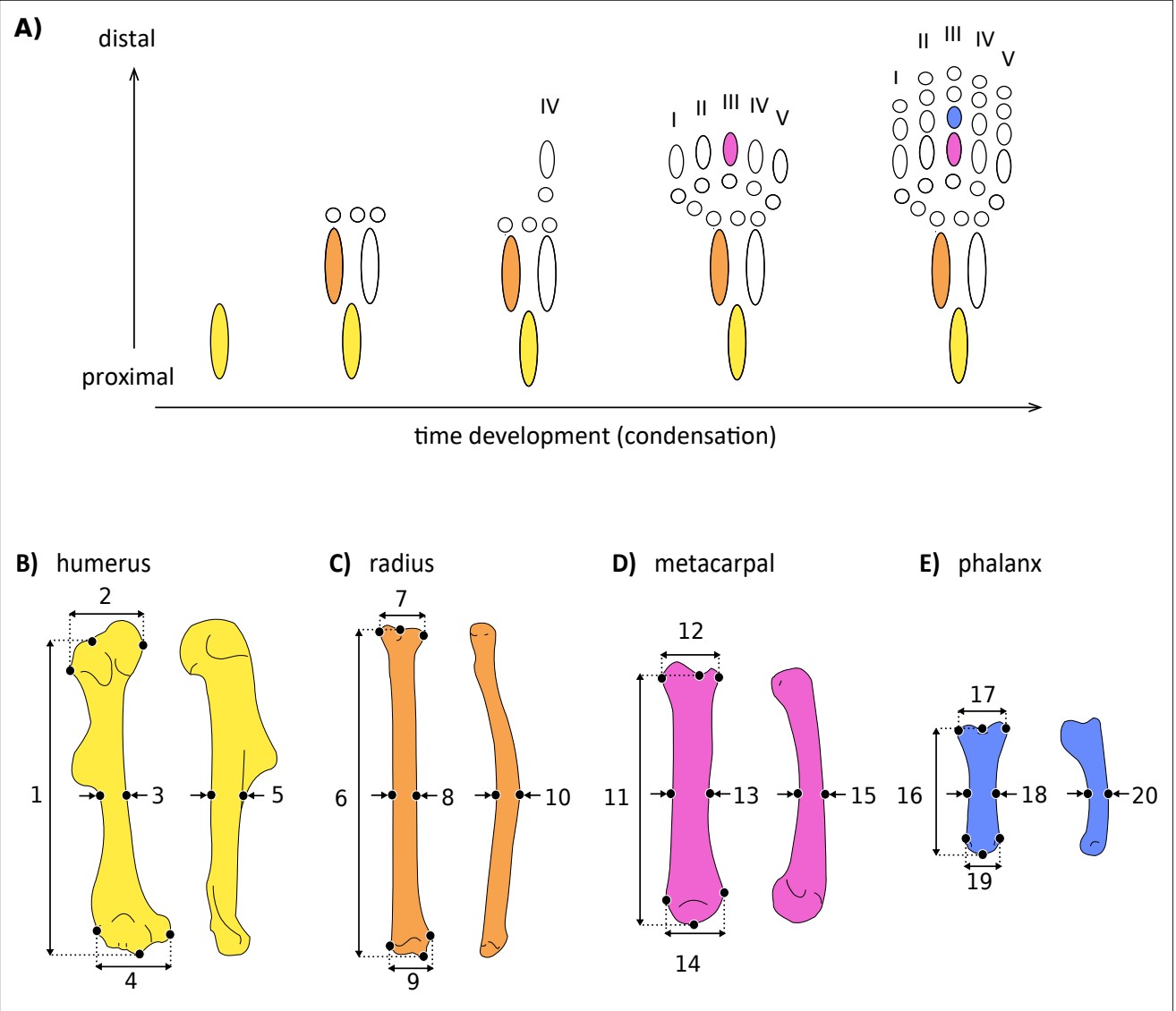

**Figure 2.** Simplified scheme of the developmental sequence of limb condensation (**A**), indicating the bones analysed and the linear measurements obtained. (**B**) Humerus in anterior (right) and lateral (left) view: (1) length, (2) proximal width, (3) mid-shaft width, (4) distal width, and (5) height. (**C**) Radius in anterior (right) and lateral (left) view: (6) length, (7) proximal width, (8) mid-shaft width, (9) distal width, and (10) height. (**D**) Third metacarpal in dorsal (right) and lateral (left) view: (11) length, (12) proximal width, (13) mid-shaft width, (14) distal width, and (15) height. (**E**) First phalanx of the digit III in dorsal (right) and lateral (left) view: (16) length, (17) proximal width, (18) mid-shaft width, (19) distal width, and (20) height. Detailed description of each measurement can be found in *Supplementary file 2a*.

what extent is the temporal structure of proximo-distal bone condensation consistent with the macro-evolution of limb segment morphologies. Mammals are an ideal group to address this question given their exceptional morphological and ecological diversity, combined with a substantial literature on the functional variation and the evolutionary development of their limbs (*Figure 1*; *Chen and Wilson, 2015*; *Grossnickle and Newham, 2016*; *Howenstine et al., 2021*; *Lungmus and Angielczyk, 2021*; *Maier et al., 2017*; *Polly, 2007*; *Sears et al., 2007*; *Weisbecker, 2011*; *Weisbecker et al., 2008*). We examined the diversification of limb skeletal elements by quantifying morphological diversity and integration using linear measurements of four forelimb bones (*Figure 2B–E*, *Supplementary file 2a*). We also estimated the macroevolutionary patterns of these elements using multivariate phylogenetic comparative methods. First, we quantified the morphological diversity of each segment, testing the hypothesis that distal bones are morphologically more diverse than the proximal structures as is predicted by development. Next, we investigated whether the strength of within-element integration

differs between proximal and distal limb elements. We predicted that proximal elements would be more integrated than distal ones, due to their earlier condensation during development. Finally, we inferred the macroevolutionary patterns for bones belonging to all limb segments, predicting positive associations between the temporal sequence of bone condensation and the capacity for evolution to generate morphological diversity. To our knowledge, this is the first time that the evolutionary patterns observed in the form of proximal versus distal limb elements are investigated using a broad phylogenetic and ecological sample of mammals, essential to address these questions.

## Results

### Morphological diversity

Among the three different evolutionary models examined (Brownian motion, Early-Burst and Ornstein-Uhlenbeck), the Ornstein-Uhlenbeck (OU; see *Hansen, 1997*) process is the one that better predicts the pattern of evolution for all bones measured (*Supplementary file 2b*). We inferred morphological diversity for each bone using the determinant and the trace of the original dataset (*Supplementary file 2c*) and of simulated trait matrices. Determinants and traces of matrices offer different but complementary generalized metrics to describe the variation of multidimensional data. The matrix trace provides information about the accumulated trait variance, whereas the determinant provides information about the volume occupied by the multivariate data. Both show similar patterns, in which morphological variation increases along the proximo-distal axis, consistent with the timing of limb condensation during development (*Figure 3A and B*). The early-condensing humerus is the least variable structure (determinant = 0.0015, trace = 0.0079), and the late-condensing phalanx is the most diverse element measured (determinant = 0.0019, trace = 0.0135), followed by the third metacarpal (determinant = 0.0017, trace = 0.0101, *Supplementary file 2c*). All pairwise comparisons between elements are significant (*Table 1*), although the differences of the determinant distributions of the radius and the metacarpal (p=0.017) are smaller than when using the trace results (p<0.001).

### Phenotypic integration

Integration, inferred here by the values of eigenvalue dispersion, is stronger for distal elements compared to proximal ones, the phalanx being the most integrated element, followed by the metacarpal (*Figure 3A and C*). The values of integration do not progressively increase along the proximo-distal axis. Instead, the radius is the least integrated structure, and the more proximal humerus is the second least integrated trait. All pairwise comparisons between elements are significant (*Table 1*).

### Stationary variances

Traits evolving under an OU process change at a given step variance ($\sigma^2$) with a strength of constrains ($\alpha$) towards an adaptive optimum ($\theta$) (*Hansen, 1997*). We interpreted the tempo of evolution of traits considering the mean stationary variance ($\sigma^2/2\alpha$) of each bone, which is a measure of rate under the OU process (*Hunt, 2012*). The stationary variance, referred here as evolutionary lability, represents the expected variation when the OU process is at equilibrium (i.e. around the optimum): the higher the stationary variances, the greater – or more labile – is the phenotypic change around the trait optimum (see *Friedman et al., 2021*; *Gearty et al., 2018*; *Hansen, 1997*; *Joly et al., 2018*; *Weaver and Grossnickle, 2020*). The stationary variances are significantly higher for distal elements compared to proximal ones. The metacarpal shows the highest stationary variance, followed by the phalanx (*Figure 3A and D*). There are no significant differences in the stationary variances at which the humerus and the radius evolve, these values being significantly lower than those of the two autopodial elements (*Table 1*). Thus, whereas these results are in line with the predictions of the developmental hypothesis in showing greater evolutionary lability in the distal elements, they do not support the idea of a proximo-distal gradient of increasing stationary variances.

## Discussion

The remarkable diversity of limb morphologies seen in mammals reflects the rich ecological and functional diversity that has evolved in this group (*Polly, 2007*). However, such outstanding morphological variation does not evolve uniformly among segments. Based on linear measurements of limb bones

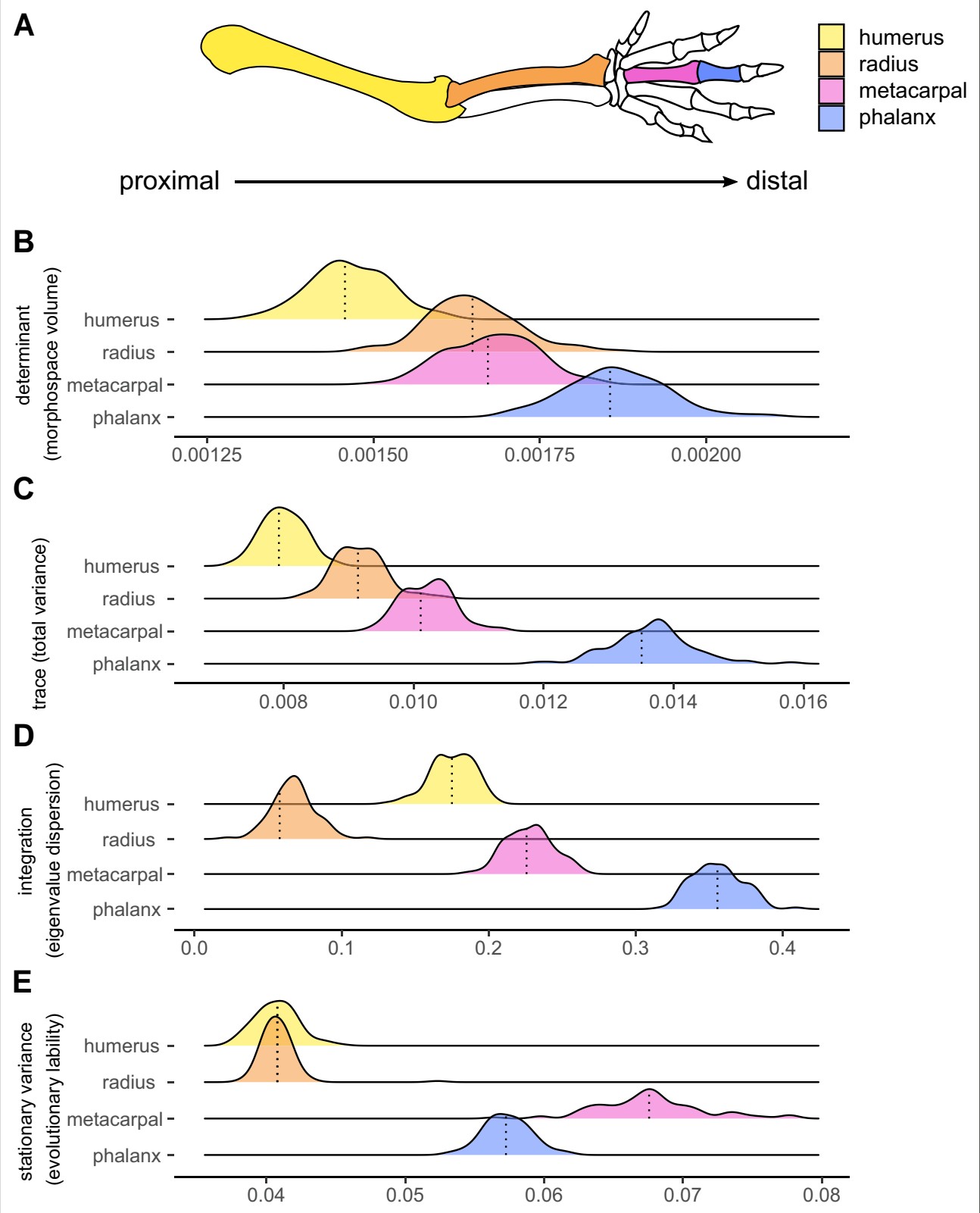

**Figure 3.** Components of the morphological evolution of forelimb skeletal elements. (**A**) Forelimb schematic, with colours indicating bones along the proximo-distal axis: the humerus (yellow), radius (orange), third metacarpal (pink), and the first phalanx of digit III (blue). Reference lines indicate empirical values (**B, C and D**) or the median values (**E**) from 100 different topologies. replicated (**B**) Morphological diversity of limb bones inferred by matrix determinant. (**C**) Morphological diversity of limb bones, inferred by matrix trace. (**D**) Trait integration. (**E**) Stationary variance.

**Table 1.** Limb bone pairwise comparison of integration, determinant, trace, and stationary variance computed by a Tukey Test following an ANOVA.

Pairwise differences (Diff) of each metric are indicated with the lower (Lwr) and upper (Upr) 95% CI, as well as the adjusted P-values. Hum = Humerus, Rad = Radius, Met = Metacarpus and Phal = Phalanx.

|  |  | Rad-Hum | Met-Hum | Phal-Hum | Met-Rad | Phal-Rad | Phal-Met |
|---|---|---|---|---|---|---|---|
| Determinant | Diff | 1.8E-04 | 2.1E-04 | 4.0E-04 | 2.9E-05 | 2.2E-04 | 1.9E-04 |
|  | Lwr | 1.6E-04 | 1.9E-04 | 3.8E-04 | 3.8E-06 | 1.9E-04 | 1.6E-04 |
|  | Upr | 2.1E-04 | 2.4E-04 | 4.3E-04 | 5.3E-05 | 2.4E-04 | 2.1E-04 |
|  | P-value | <0.001 | <0.001 | <0.001 | 0.017 | <0.001 | <0.001 |
| Trace | Diff | 1.2E-03 | 2.2E-03 | 5.7E-03 | 1.0E-03 | 4.5E-03 | 3.4E-03 |
|  | Lwr | 1.0E-03 | 2.1E-03 | 5.5E-03 | 8.8E-04 | 4.3E-03 | 3.3E-03 |
|  | Upr | 1.4E-03 | 2.4E-03 | 5.8E-03 | 1.2E-03 | 4.6E-03 | 3.6E-03 |
|  | P-value | <0.001 | <0.001 | <0.001 | <0.001 | <0.001 | <0.001 |
| Integration | Diff | −0.108 | 0.052 | 0.180 | 0.160 | 0.288 | 0.128 |
|  | Lwr | −0.114 | 0.047 | 0.175 | 0.155 | 0.283 | 0.122 |
|  | Upr | −0.102 | 0.058 | 0.186 | 0.166 | 0.294 | 0.133 |
|  | P-value | <0.001 | <0.001 | <0.001 | <0.001 | <0.001 | <0.001 |
| Stationary variance | Diff | 0.000 | 0.027 | 0.017 | 0.027 | 0.017 | −0.010 |
|  | Lwr | −0.001 | 0.026 | 0.016 | 0.026 | 0.016 | −0.011 |
|  | Upr | 0.001 | 0.028 | 0.017 | 0.028 | 0.017 | −0.009 |
|  | P-value | 0.989 | <0.001 | <0.001 | <0.001 | <0.001 | <0.001 |

we show a general pattern of morphological diversity in Mammalia in which distal elements such as phalanges and metacarpals are in general more disparate and show greater evolutionary lability, as indicated by our measures of stationary variance, than more proximal elements such as the humerus and radius. These results are consistent with the hypothesis that the among-species diversity of limb element morphologies is predicted by the timing of element condensation during development. Conversely, developmental constraints imposed by early versus late morphogenesis do not seem to determine differences in within-bone integration; we found that the latest-condensing elements of the hand are more integrated than the earlier-condensing humerus and radius. We hypothesize that the degree of functional specialization across segments might play a role on the levels of within-element integration, with the autopod potentially being more specialized and therefore exhibiting greater integration. We further show that distal elements evolve, on average, with greater stationary variances (i.e. faster) than the proximal limb elements.

## Limb segments: a proximal to distal gradient of increasing diversity

Previous studies have described the exceptional meristic variation in the autopod in contrast with the proximal and intermediate limb (*Holder, 1983*). Here, we confirm that such diversity is also detected in the form of hand bones. A pattern of increased morphological diversity along the proximal-distal limb axis is consistent with the prediction that lower proximal diversity might have been driven by developmental canalization (*Hallgrímsson et al., 2002*). A similar pattern has been previously documented for anurans, in a study that compared shape variation of the humerus and the radioulna of the forelimb, as well as the femur, the tibiofibula, and the tarsus of the hind limb (*Stepanova and Womack, 2020*). Although this study did not include the digital elements of the hand (metacarpals and phalanges), it showed that late developing distal structures are not only more diverse but also evolve faster than the most proximal elements belonging to the stylopod (*Stepanova and Womack, 2020*). Microhabitat use also explains more shape variation in the distal elements compared to the most proximal limb bones, suggesting that functional specialization evolves differently along the

proximo-distal limb axis (*Stepanova and Womack, 2020*). Combined with our results, these findings provide evidence that the proximal-distal gradient of variation in limb structures may consist of an early conserved pattern shared across tetrapods, supporting the hypothesis that the timing of development affects the intrinsic capacity of an organism to generate variation and facilitate subsequent functional diversification.

Functional variation is often a good predictor of the pattern of morphological variation of limb bones (*Chen and Wilson, 2015*; *Fabre et al., 2013*; *Grossnickle et al., 2020*; *Weaver and Grossnickle, 2020*). The autopod is the structure that interacts directly with the surrounding environment, performing important activities such as providing support to the body during locomotion and, in some cases, digging, handling food, grooming, and mediating social interactions (*Biewener and Patek, 2018*; *Mc Grew et al., 2001*; *Naghizadeh et al., 2020*; *Sustaita et al., 2013*; *Weisbecker and Warton, 2006*). Our results corroborate the idea that the hand bones are subjected to more dynamic selective pressures that ultimately favour greater diversity and evolutionary lability compared to proximal segments. However, few studies have quantified the functional relationships driving autopod variation in mammals (*Almécija et al., 2015*; *Rolian, 2009*; *Weisbecker and Schmid, 2007*; *Weisbecker and Warton, 2006*). As the number of digits, and the number of phalanges in each digit, varies within most groups of tetrapods, including mammals, autopodial morphology is difficult to quantify in a comparable, homologous way among species. Although our data support this association, we do not explicitly test for the relationship between the variation observed in each bone and its degree of functional specialization. Further investigations are needed to quantify the impact of these parameters on limb diversification and need to set up a priori testable hypotheses.

To our knowledge, our study comprises the most comprehensive taxonomic dataset on the forearm morphology of mammals. The use of linear measurements succeeded at establishing comparable topological distances and provided a robust overview for the global morphological diversity between limb segments across the mammalian tree of life. However, this method imposes some limitations on capturing detailed local shape variation. For example, the proximal joint at the humerus can encompass complex surfaces which determine the mobility of the limb (*Arias-Martorell, 2019*; Veeger and *Veeger and van der Helm, 2007*), but variation therein is not captured by our measurements. Likewise, the shape and size of the deltopectoral crest of the humerus may display considerable interspecific variation (*Chen and Wilson, 2015*; *Hopkins and Davis, 2009*; *Samuels and Van Valkenburgh, 2008*), and is also not quantified here. Similar structures are not present at the joints or at the longitudinal surfaces of the phalanges. Thus, it is not clear whether adding such morphological features would have resulted in an increase of morphological diversity in proximal bones compared to the distal ones. Either way, previous studies that have incorporated complex geometric surfaces of the long bones detected that the robustness (i.e., the correlation of length and thickness) is one of the principal factors contributing to the pattern of morphological variation (*Fabre et al., 2017*; *Fabre et al., 2014*; *Michaud et al., 2020*), consistent with our results.

Having the zeugopod solely represented by the radius might as well have obscured some of the diversity present in the intermediate limb segment. The ulna is highly variable, with the olecranon particularly being a strong predictor of locomotor habit (*Chen and Wilson, 2015*; *Lungmus and Angielczyk, 2021*; *Milne and Granatosky, 2021*; *Samuels and Van Valkenburgh, 2008*; *Van Valkenburgh, 1987*). However, due to the high variation of the ulna, the topological distances used to describe the skeletal morphology (length, width, and height) cannot be applied to this bone in all species as the ulna is distally reduced or fused to the radius in many taxa (*Sears et al., 2007*), thus preventing us from quantifying the diversity between this and other limb bones. We encourage future studies to include the ulna and to use geometric morphometrics of the joints to complement our findings with detailed information on shape variation across limb segments.

## Functional predictors of bone integration

The high integration detected in the mammalian hand suggests that developmental constraints of early versus late bone condensation do not predict within-element covariation. These findings do not imply that development is unimportant for the individual integration of elements. Yet, in the matter of relative bone integration, the timing of condensation is unable predict which elements are the most and the least integrated. The proximal and distal humeral joints perform different functions and allow very different movements: the proximal head connects the limb to the pectoral girdle at the shoulder

through a complex ball-and-socket articulation (*Arias-Martorell, 2019*; Veeger and *Veeger and van der Helm, 2007*), and it distally articulates with the ulna and the radius at the elbow driving forelimb mobility and stability (*Fabre et al., 2014*). The radial joints are also involved in different functions, having a proximal head connected to the elbow and an enlarged distal extremity articulated at the wrist with carpals and sometimes the ulna (*MacLeod and Rose, 1993*; *Polly, 2007*). Because they are involved in different functions, the articular surfaces of long bones are differently impacted by functional specialization related to locomotor habit (*Fabre et al., 2014*; *Lungmus and Angielczyk, 2021*; *MacLeod and Rose, 1993*). In terms of within-element integration, the different functional demands at the proximal and distal bone extremities might encompass a reduction of covariation between these traits, as detected here for the humerus and the radius. The metacarpal and the phalangeal articulations, on the other hand, work more similarly: phalanges articulate with the metacarpals at a bi-axial-joint (movement at two axes: flexion/extension, abduction/adduction) and articulate with each other at hinge joints which allow only one axis of movement (flexion and extension; *Napier and Tuttle, 1993*). The strong integration of hand bones detected for mammals indicates that these elements experience a highly correlated evolution, which in this case might also emerge from functional similarity and interdependence at the articulations.

## Integration and evolutionary lability

The relationship between integration and morphological variation is not always consistent among traits and taxa (*Felice et al., 2018*). Whereas some studies have shown clear positive associations between high integration and phenotypic variation (*Fabre et al., 2021*; *Fabre et al., 2020*; *Randau and Goswami, 2017*), negative associations have been also reported (*Felice and Goswami, 2018*; *Goswami and Polly, 2010*). We find no evidence for a strong correspondence of integration with morphological diversity in proximal forelimb segments: the radius exhibits greater diversity of form than the humerus but presents the weakest values of integration among the bones measured. For the distal elements, however, our results show that the highly integrated autopod, especially the phalanx, also corresponds to the most diverse structure of the limb (*Figure 3*). These differences might reflect how selection interacts with the intrinsic and extrinsic constraints on variation. Though integration may constrain the evolution of the phenotype to a limited portion of morphospace, it may also promote variation by driving the evolution of these traits in response to selection for functional specialization (*Felice et al., 2018*; *Goswami et al., 2014*; *Hansen and Houle, 2008*; *Lande, 1979*). Such dynamics appear to be observed in the distal elements: high integration in the phalanx and metacarpus, possibly favoured the evolution of functionally specialized autopod structures, contributes to the high variation observed in mammalian hand bones. Future studies will benefit from including extinct taxa to understand how morphological diversity and integration of limb bones evolved in deep time. Such analyses would further provide insights into whether these patterns are consistent between major taxonomic and ecological groups and through time and would provide information on when they first appeared during mammalian evolution.

## Evolutionary lability of the autopodal elements: functional associations

The autopodial bones evolve, on average, with greater stationary variances around their optima than the stylopod and the zeugopod (*Figure 3*). Although the developmental hypothesis predicted that the fastest evolving structures would belong to the late-condensed distal limb, evolutionary lability do not increase in a perfect proximal-to-distal pattern, and the third metacarpal is the structure with the highest stationary variances, followed by the phalanx. These findings suggest that functional selection (resulting from the direct impact of autopodial structures on locomotor performance) combined with the higher potential of development to generate variation in the morphology of more distal limb elements, facilitated the evolution of high autopodial disparity in response to varying environmental demands across mammals. Although this subject remains largely unexplored, some studies provide cues about the possible association of function with the evolutionary lability of the autopod morphology in tetrapods (*Ledbetter and Bonett, 2019*; *Rolian, 2009*).

Notable transformations in the metacarpal and phalangeal morphology are observed in cursorial taxa that present specializations allowing for endurance running, typically involving the elongation of the distal limb in relation to proximal segments (*Polly, 2007*). These transformations may explain part of the results observed in our study. For example, morphological adaptations to cursoriality mostly

encompass the modification of autopod posture to digitigrady (animals that stand on the distal ends of metapodials and middle phalanges, such as cats and dogs) and unguligrady (animals that stand on their hooved distal-most phalanx, such as horses and cows; *Clifford, 2010*; *Polly, 2007*; *Wang, 1993*). Digitigrady is observed in many carnivorans providing limb elongation and thus increasing stride length (*Polly, 2007*; *Wang, 1993*). Extant horses exhibit one of the most dramatic modifications of the third metapodial and phalanges among all unguligrade taxa: the limb is uniquely supported by the third toe, which is considerably enlarged and elongated, whilst the lateral fingers are markedly reduced (*McHorse et al., 2019*). One recent study suggested that the evolutionary transitions in foot and hand postures are associated with strong selection for rapid changes in increasing body size (*Kubo et al., 2019*). Although a digital posture presumably implies morphofunctional specialization of the distal limb, it is not clear if the acceleration of body mass evolution during autopod posture transitions has also affected the rates of morphological change of the hand and foot. Autopodial specialisations are also evident among smaller-sized mammals. For example, body size is positively associated with the tempo of evolution of postcranial morphology (hand and foot bones included) in both ground and tree dwelling animals, where medium-sized animals tend to exhibit higher stationary variances than small-sized species (*Weaver and Grossnickle, 2020*). Overall, these examples suggest that functional specializations related to the locomotion and size likely played a role in driving the morphological evolution of the limb, potentially driving the accelerated evolution of hand bone morphology. Further investigations are needed to better understand the associations of body size and functional variation with the evolutionary dynamics of limb diversification.

## Conclusion

This study uses a macroevolutionary framework to compare, for the first time, the general patterns of form diversification of proximal and distal limb elements in mammals. Our results reveal that the evolution of the mammalian forelimb involves different patterns of morphological diversification when comparing limb segments along a proximal–distal gradient. We detected that the diversification of autopodial elements was much more dynamic than that of the zeugopod and stylopod, involving higher morphological diversity, stronger integration, and greater evolutionary lability at distal structures. Specifically, we corroborate the premise that the late-condensing distal elements such as metacarpals and phalanges (in the autopod) exhibit higher morphological diversity than early-condensing, more proximal, elements. This pattern might emerge from different levels of constraints during the developmental succession. Yet, no proximo-distal gradient in stationary variance was observed. Furthermore, such temporal constraints of development do not explain the patterns of limb evolution alone, as functional specializations also play an important role on the diversification of the forelimb. Particularly, the strong integration of the autopodial elements most likely reflects the functional similarity and interdependence between joints in response to functional demands. We highlight the importance of considering variation induced by development to understand the macroevolutionary outcome of adult morphologies, and we hope that these results will contribute to better understand the association of limb segment variation and ecological diversity.

## Materials and methods
### Taxonomic sampling and data acquisition

We sampled 638 species of mammals (670 specimens), representing 598 genera of 138 living families (*Figure 1*). Sampling varies from one to four individuals per genus. We provided micro-CT-scans and surface scans of 58 small to medium sized-specimens from different institutions (available online at MorphoSource.org, *Supplementary file 1*), 23 of them previously used by *Martín-Serra and Benson, 2020*. The digital dataset was combined with 351 meshes available on MorphoSource.org (*Supplementary file 1*). Image stacks were converted into three-dimensional models using Avizo 8.1.1 (1995–2014 Zuse Institute Berlin), where scale dimensions were incorporated based on the voxel size of each scan. Data collection from the digital models was also conducted in Avizo 8.1.1 (1995–2014 Zuse Institute Berlin). We complemented this dataset with measurements provided by caliper of 261 medium to large body-sized species from the mammal collection of the Muséum National d'Histoire Naturelle (Paris, France; *Supplementary file 1*).

We measured 20 linear distances from anterior limb bones, including the humerus, the radius, the third metacarpal and the first phalanx of digit III. We acquired five measurements for each element: length, widths (proximal, mid-shaft and distal) and height (*Figure 2*, see detailed description in *Supplementary file 2a*). We opted not to include the ulna because this bone is fused to the radius in many taxa (see *Sears et al., 2007*), preventing the acquisition of such measurements. The metacarpal and first phalanx of digit III were sampled because this is the only digit present in the hands of all mammalian lineages, even in groups that exhibit digit loss or fusion with other autopodial elements, such as in golden moles and ungulates (*Clifford, 2010*; *McHorse et al., 2019*; *Prothero, 2009*). Each individual was measured twice with the subsequent calculation of the mean and standard error in order to verify measurement error. The error estimate was most often below 1.5% regardless of an animal's size and the measurement method, demonstrating consistency and repeatability of the methods employed. Body mass values were rarely available for the individuals measured, so we assembled the average species body mass of adults from the PanTHERIA database (*Jones et al., 2009*) and complemented by literature sources when necessary (*Supplementary file 1*). When species level was not identified, we used the mean body mass available for the genus. Species taxonomy followed the Mammal Diversity Database published by *Burgin et al., 2018*.

## Comparative analyses

Analyses were implemented in R 4.1.2 (*R Core Team, 2021*). We used the phangorn R package (*Schliep, 2011*) to estimate a maximum clade credibility (MCC) tree from a posterior sample of 10,000 trees published by *Upham et al., 2019*. Because the incorporation of some species was available only at the genus level, we pruned the MCC tree to genus level, according to the taxa sampled by our study, and calculated the genus mean per trait whenever we had more than one specimen measured per genus.

Allometry generally explains most of morphological variation, as body parts usually grow together, masking variation mediated by local development (*Marroig, 2007*; *Raff, 1996*). Because we are particularly interested in understanding morphological constraints imposed by the local development of the limb, we decided to remove the allometric component of our dataset in order to reduce variation associated with other sources of development. We could not retrieve the individual body masses for most of the species included, so we calculated geometric means as a proxy for body size by including values of the individuals themselves and the average species body mass. First, we transformed body mass into linear scale by taking the cube root prior to log10-transformation (*Harmon et al., 2010*). We calculated the geometric means of all measurements acquired, including the linear scaled body mass, and then we fitted the log10-transformed trait means in a phylogenetic generalized least-squares (PGLS) using the geometric means as a predictor. We grouped the traits by bone and fitted the linear models for each skeletal unit with mvgls() function from mvMORPH R package (*Clavel et al., 2019*; *Clavel et al., 2015*). We calculated the fit of three models of evolution using LASSO penalization: Brownian Motion (BM), Ornstein-Uhlenbeck (OU), and Early Burst (EB). We compared the likelihood of the model fits with Generalized Information Criterion (GIC) to establish which model provided the best fit.

The OU model of evolution had the best fit for all the linear regressions accounting for the geometric means using the MCC tree (*Supplementary file 2b*). To evaluate whether using the species average value (and not the individual body mass) would bias the results, we performed supplemental PGLS removing the average body mass from the geometric means. The results between bones remained the same (*Supplementary file 2c and d*), so we maintained the body mass in the geometric means for the downstream analyses. We used a parametric bootstrap approach to assess the uncertainty around point estimates for morphological diversity and integration. We first simulated 100 datasets for each bone on MCC tree using the OU process fit (that is, the best fit model on our original data) with parameters estimates from the empirical regression (function mvSIM() from mvMORPH; *Clavel et al., 2015*; *Clavel et al., 2019*). The model (the body size PGLS under an OU process) was then fit to these 100 simulated traits, and the distribution of parameters estimates obtained was used to assess the variability around the point estimate (for the determinant, the trace, and the measure of integration) obtained on empirical data.

## Morphological diversity and phenotypic integration

Morphological diversity for each bone was interpreted as the values of the determinant and the trace of simulated matrices. The trace is the sum of the diagonal elements of the trait covariance matrix, that is, the sum of individual traits variance (sum(diag()), **R Core Team, 2021**). The determinant is a scalar measure that summarize the information contained in a square matrix (det(), **R Core Team, 2021**). For a covariance matrix, it corresponds to a generalized measure of variance, because contrary to the trace, the determinant account for the correlations/covariances between the traits (**Rencher, 2002**). We scaled the determinants by transforming their absolute value to the power of one divided by five, which is the number of dimensions of each matrix (i.e. the number of measurements). Differences in the determinant and trace between skeletal elements were evaluated by ANOVA followed by Tukey Tests (function TukeyHSD() from stats R package) of the 95% confidence interval (CI).

We calculated the magnitude of integration for each bone separately, based on eigenvalue dispersion in their respective matrices. We transformed the simulated covariance matrices into correlation matrices and provided integration values as the standard deviation of eigenvalues relative to their theoretical maximum (**Haber, 2011**; **Pavlicev et al., 2009**). We calculated the integration as the dispersion of the standard deviation of eigenvalues of our trait matrices, following **Pavlicev et al., 2009**. For instance, highly integrated traits have most of the independent variance concentrated in the first few eigenvalues, while uncorrelated traits have the variance similarly distributed between eigenvalues (**Pavlicev et al., 2009**). Eigenvalue dispersion was inferred from CalcEigenVar() function of evolqg R package (**Machado et al., 2019**; **Melo et al., 2015**), which calculates the relative eigenvalue variance of the matrix as a ratio between the observed variance and the theoretical maximum for a matrix of the same size and trace (**Machado et al., 2019**). Differences between distributions were computed by an ANOVA and detailed by Tukey Tests of the 95% CI.

## Macroevolutionary patterns

Finally, we were interested in estimating the tempo of evolution of each bone. To assess variability due to the tree topology and branching times uncertainties, we replicated the body mass linear regressions with 100 randomly sampled trees from **Upham et al., 2019**. We fitted these linear regressions under an OU process and estimated the average rates of evolution ($\sigma^2$) per bone. Under a Brownian motion process, the tempo of evolution can be directly inferred from the $\sigma^2$, which represents the total variance of traits changes linearly, as a function of the traits covariances matrix and time (**Harmon, 2019**). In an OU process, however, traits evolve towards an optimum θ with an attraction α. The main difference between BM and OU, is that the trait variance changes with time in BM, while it is not related to time in OU when stationary (**Hunt, 2012**). Assuming that time was long enough in an OU process (so that the process is stationary, e.g., reached the optimum), its covariance matrix, equivalent to a BM matrix of traits variance and covariance, depends only on the parameters $\sigma^2$ and α (**Hunt, 2012**). A comparable rate metric for traits evolving under OU process is then the stationary variance ($\sigma^2/2\alpha$), representing the variance of traits distribution per evolutionary steps (or the variance of traits when lineages were given enough time to reach their optima and the process is in equilibrium; **Hansen, 1997**; **Hunt, 2012**). Therefore, we calculated the mean stationary variance of bones from the matrices fitted under OU process (function stationary() from mvMORPH; **Clavel et al., 2015**). We compared their distributions using ANOVA followed by a 95% confidence interval Tukey Test.

## Source code

Data and codes will be made available on Dryad Digital Repository upon to manuscript publication.

## Acknowledgements

We thank the following collection managers at the MNHN, Paris, for their support during data acquisition: Alexander Nasole, Aude Lalis, Aurélie Verguin, Céline Bens, Géraldine Veron, Jacques Cuisin, Joséphine Lesur and Violaine Colin. Creation of datasets accessed on MorphoSource was made possible by the following funders and grant numbers: NSF DBI-1701713, 1701714, 1701737, 1702263, 1701665, 1701767, 1701769, 1701870, 1701797, 701851, 1702442, 1902105, BCS 1317525, BCS 1540421, BCS 1552848; ERC-2015-STG-677774 (TEMPO Mammals project to RB) and Leakey Foundation. Project funding for digital acquisition of each used specimen is detailed in *Supplementary*

*file 1*. We also thank Anjali Goswami, Helder Gomes Rodrigues, Eric Guilbert and Loïc Kéver for their insightful comments during the elaboration of this study. This work was supported by CNPq doctoral grant to PSR (process #204841/2018–6).

## Additional information

### Funding

| Funder | Grant reference number | Author |
|---|---|---|
| Conselho Nacional de Desenvolvimento Científico e Tecnológico | 204841/2018-6 | Priscila S Rothier |
| European Research Council | 2015-STG-677774 | Roger BJ Benson |

The funders had no role in study design, data collection and interpretation, or the decision to submit the work for publication.

### Author contributions

Priscila S Rothier, Conceptualization, Data curation, Formal analysis, Funding acquisition, Validation, Investigation, Visualization, Methodology, Writing - original draft, Project administration, Writing – review and editing; Anne-Claire Fabre, Conceptualization, Methodology, Writing – review and editing; Julien Clavel, Formal analysis, Validation, Methodology, Writing – review and editing; Roger BJ Benson, Data curation, Writing – review and editing; Anthony Herrel, Conceptualization, Supervision, Funding acquisition, Methodology, Project administration, Writing – review and editing

### Author ORCIDs

Priscila S Rothier ![ORCID] http://orcid.org/0000-0003-3017-6528

### Decision letter and Author response

Decision letter https://doi.org/10.7554/eLife.81492.sa1
Author response https://doi.org/10.7554/eLife.81492.sa2

## Additional files

### Supplementary files

• Supplementary file 1. Analyzed specimens, indicating taxonomy, institutional identification, method of morphometric acquisition, MorphoSource ID and associated projects (when applicable) and body mass (grams, from PanTHERIA or indicated literature).

• Supplementary file 2. Trait description and model fit. (a) Linear measurements obtained for each specimen extracted either by landmark positioning in 3D digital models or by direct caliper measurements (illustrated in *Figure 2*). (b) Fits of linear models of evolution for each bone, highlighting in bold the best model fitted according to generalized information criterion (GIC) and loglikelihood (logLik). $\sigma^2$=mean evolutionary rate, $\alpha$=attraction toward optimum, stat.var.=mean stationary variance. (c) Empirical values from PGLS regression computing for body mass in the geometric means. (d) Empirical values from PGLS regression without body mass in the geometric means.

• MDAR checklist

### Data availability

Morphometric data and R codes are available on Dryad (https://doi.org/10.5061/dryad.0cfxpnw6h).

The following dataset was generated:

| Author(s) | Year | Dataset title | Dataset URL | Database and Identifier |
|---|---|---|---|---|
| Rothier PS, Fabre A, Clavel J, Benson R, Herrel A | 2023 | Input data from: Mammalian forelimb evolution is driven by uneven proximal-to-distal morphological diversity | https://doi.org/10.5061/dryad.0cfxpnw6h | Dryad Digital Repository, 10.5061/dryad.0cfxpnw6h |

The following previously published dataset was used:

| Author(s) | Year | Dataset title | Dataset URL | Database and Identifier |
|---|---|---|---|---|
| Upham N, Esselstyn J, Jetz W | 2019 | Inferring the mammal tree: Species-level sets of phylogenies for questions in ecology, evolution, and conservation | https://dx.doi.org/10.5061/dryad.tb03d03 | Dryad Digital Repository, 10.5061/dryad.tb03d03 |

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
