## [Editor Report]

This study reports an interesting analysis of evolutionary variation in forelimb/hand bone shapes in relation to functional and developmental variation along the proximo-distal axis. The authors found expected and compelling patterns of evolutionary shape variation along the proximo-distal axis but less expected, yet equally compelling, patterns of shape integration. This paper will be of interest to researchers working on macroevolutionary patterns and sources of morphological diversity.

---

## [Decision Letter]

**Decision letter after peer review:**

Thank you for submitting your article "The mammalian forelimb diversity as a morphological gradient of increasing evolutionary versatility" for consideration by *eLife*. Your article has been reviewed by 3 peer reviewers, and the evaluation has been overseen by a Reviewing Editor and Marianne Bronner as the Senior Editor. The following individuals involved in the review of your submission have agreed to reveal their identity: David M. Grossnickle (Reviewer #2); Francois Gould (Reviewer #3).

Essential revisions:

The reviewers make sensible requests for major revisions to the article, mostly concerning the need to include essential information that is missing from the methods and discussion, and to give more serious consideration to other (functional, allometric) explanations for the patterns that are presented. Since most of the suggested revisions are text-based rather than data/methods-based, it should be straightforward to address them.

*Reviewer #1 (Recommendations for the authors):*

Abstract:

Although the authors tempered their claims around decisive support for developmental constraints as the sole driver of limb morphological diversity, the abstract conclusion still overstates the results. The line "suggesting that strong functional selection, combined with the higher potential of development to generate variation in more distal limb structures, facilitate the evolution of high autopodial disparity in mammals" overstates the conclusions that can be drawn from these data. We can't be sure that stronger functional selection does not solely generate the variation in distal limb structures. Nor can we be sure the pattern is not simply a result of the differing developmental potential for variation. The combination of stronger functional selection with the higher potential of development to generate variation is a really interesting hypothesis but it is not necessarily suggested by the data. These are simply possible explanations for the patterns described in the paper.

Introduction:

The introduction primarily cites and discusses mammalian limb studies, despite the relevance of the suggested findings beyond mammals and forelimbs. Furthermore, relevant work exists in other tetrapod clades and structures related to evolutionary rates of later-developing traits and proximo-distal variation.

Results:

When explaining stationary variances you state "which represents the expected variation when the process is in equilibrium and summarizes the relative influence of stochastic factors in evolution". It would be really helpful to explicitly say how this measure summarizes the relative influence of stochastic factors by stating something like "a lower stationary variance value means X whereas a higher stationary variance means Y".

Discussion:

Line 235: rephrase "or both in combination therefore".

Line 327: space needed before citation "species(Weaver and Grossnickle, 2020)"

Finally, although the evolutionary patterns here are interesting there is a missed opportunity with this extensive sampling to test for effects of ecology, locomotion, and/or digit number variation on the patterns observed. This would have provided more clear evidence (or lack of) for some of the suggestions in the discussion.

Methods:

Although all bones can be relatively scaled, do you think differences among bones in their original size and shape, and thus, differences among bones in the ultimate spacing of the landmarks, could affect the variation in shape seen among structures? Specifically, a smaller, wider bone when scaled equally to a long thin bone might create really different potentials for measured shape change when you think about how those landmarks will vary across different bone shapes in each case. If this influences the analysis/results, it could be a confounding factor since the bones get smaller from proximal to distal.

I found it a bit confusing how the size was accounted for in these analyses. The size was the mean mass for the genera from a database? Not from the specimens themselves? Would somehow using the size of the bones themselves provide a better proxy for the body size of the actual animal measured?

*Reviewer #2 (Recommendations for the authors):*

Line 1: In the title and main text you use the phrase "evolutionary versatility." I like it, but it's also a little vague, so it'd help if you provide your definition of "evolutionary versatility" somewhere in the text. You have the phrase in parentheses after "stationary variance" in Figure 3E, which implies that it's mostly referring to the stationary variance results, but this isn't clear in the text unless I missed an explanation. Also, you could consider replacing "evolutionary versatility" with the term "evolvability," if you think they're synonymous (e.g., see Love et al. 2021 Paleobiology, and a couple of your references that have "evolvability" in their titles). On a broader note, you often mention "constraints," but you might consider putting greater emphasis on the 'evolutionary versatility (or evolvability) of the distal elements' rather than the 'constraints on proximal elements.' Constraints are often inferred from the lack of something (negative evidence), but it's hard to tell whether it's truly constraints that have led to the lack of some morphologies. See Salazar-Ciudad (2021 Biol. Direct): "The role of development in evolution should be described in an exclusively positive way, as the process determining which directions of morphological variation are possible, instead of negatively, as a process precluding the existence of morphological variation we have no actual reason to expect." However, maybe this is all just semantics, so feel free to ignore this if you'd like.

Lines 35: You may want to explicitly state that "more integrated" refers to more within-element integration. Otherwise, it could be interpreted as between-element integration.

Line 125: You say "within-segment integration," but would "within-element" be more accurate? You report different integration results for the metacarpal and phalanx.

Line 146: Because your Methods are at the end, you may want to mention here what models were outperformed by the OU model. Otherwise, it's a little confusing to have "better" in the sentence without saying what it's better than.

Line 153: It'd be helpful for readers like me who don't have a strong stats background if you better explained "determinants" and "traces". You could do it either here or in your Methods, and/or you could cite a paper or two where they're better explained.

Line 179: Tukey's test was the post hoc test after the ANOVA, correct? It may be worth mentioning the ANOVA here.

Line 198-201: As I note in the public review, your results are likely influenced to some degree by your choice of measurements. You may want to add clarifiers to sentences like this, such as "Based on linear measurements of limb elements, we show a general.…". Or something along those lines.

Line 201: Yes, but your results are also consistent with the hypothesis that the autopod is more functionally specialized and therefore has a greater diversity of forms (and greater integration). This is an example of a spot where I recommend first discussing functional diversity because I think it offers a better explanation for your results.

Line 229: "as whales and bats" implies that if marsupials evolved to be aquatic or power-fliers then they'd be whales and bats. But those are placental groups. Maybe you mean something like "comparable to whales and bats in Placentalia".

Line 238: Pevsner et al. 2022 are cited here, but I don't see it in the References section.

Line 302-306: I really like this summary sentence – you lead with function, and then mention development, which is more in line with my interpretation of results (see comments above). I recommend moving statements like this to the forefront of the paper.

Line 384: Maybe "most part" should just be "most"?

Line 389: I like that you included body mass, but one issue is that it's from a different source (PanTheria) rather than being the weights of the actual specimens you measured, and this introduces a source of error. The masses provided in PanTheria are likely very different than the masses of the individuals you measured. If, for example, PanTheria says a species is 300g, but you measured a smaller individual that's only 100g, then this could bias all of the size-corrected measurements for that species. This is likely to have the biggest influence on smaller taxa/individuals, which can vary considerably in size even throughout a day (e.g. insectivorous bats can double in weight after a night of feeding). I doubt that this has a significant impact on your results, but I recommend justifying the use of body mass in your size-correction calculations. Also, you could go a step further and re-analyze your data without including body mass during size correction to see what influence it has on overall results.

Lines 391 and 401: Be careful when using "size" or "mass". They're not quite synonymous. On Line 391 you say "body size," but I think you mean mass. And your PGLS regressions are for correcting for size, so on Line 401 I'd say "body size PGLS" rather than "body mass PGLS".

Line 395-402: I got a little confused here. It's not immediately clear to me why you needed to simulate data. Couldn't the downstream analyses be conducted with the original dataset (or residuals after the regression) and then error or significance be calculated with the simulated datasets? Or are the simulations for ancestral state reconstructions (like SIMMAPS)? A little more information would be helpful.

Line 409: Is there a reason why you used ANOVA instead of phylogenetic ANOVA?

Figure 2: You could color-code the bones to match the colors in Figure 3.

Figure 3: Is there any way to add another part to this figure that illustrates the typical timing of condensation for different bones? It could be fairly conceptual, just showing the relative timings. I think that would help you to better emphasize/visualize your conclusions.

Figure 3D: I recommend adding a unit for integration (maybe "values of eigenvalue dispersion", Line 165) in parentheses after "Integration" in the figure and/or in the caption.

*Reviewer #3 (Recommendations for the authors):*

Given that the paper will ultimately not discuss the hindlimb at all, I am not sure the section in the introduction from lines 98 to 114 is entirely relevant. Certainly discussion of fore and limb serial homology is not relevant to the introduction of this paper

The authors mention they took multiple (two) sets of measurements of each bone to account for measurement error. However, the authors included measurements from three different sorts of data (physical specimens, microCT scans, and surface scans) over a huge size range (8 orders of magnitude from 3 g to 30 000 000 g based on the supplemental data). As such the possibility of size-related and measurement method related both relative and absolute measurement error needs to be considered. Do the authors have any numerical data, or data from published studies, addressing the issue of measurement error in large taxonomic samples including a mix of physical and digital specimens?

There are 638 species represented in the dataset, yet 670 specimens. Thus almost all species are represented by a single specimen. Indeed only 33 species in the dataset are represented by multiple specimens, and that is never more than 2. Furthermore, it appears for much of the analysis the unit of analysis is the genus mean. Thus what is the rationale for those duplicate measurements in 33 cases?

---

## [Author Response]

Reviewer #1 (Recommendations for the authors):Abstract:Although the authors tempered their claims around decisive support for developmental constraints as the sole driver of limb morphological diversity, the abstract conclusion still overstates the results. The line "suggesting that strong functional selection, combined with the higher potential of development to generate variation in more distal limb structures, facilitate the evolution of high autopodial disparity in mammals" overstates the conclusions that can be drawn from these data. We can't be sure that stronger functional selection does not solely generate the variation in distal limb structures. Nor can we be sure the pattern is not simply a result of the differing developmental potential for variation. The combination of stronger functional selection with the higher potential of development to generate variation is a really interesting hypothesis but it is not necessarily suggested by the data. These are simply possible explanations for the patterns described in the paper.

We agree with the recommendation, so we changed the sentence to emphasize that our results are consistent with the developmental hypothesis without excluding other potential drivers of variation: “High disparity of the mammalian autopod, reported here, is consistent with the higher potential of development to generate variation in more distal limb structures, as well as other possible drivers such as functional adaptation.” (lines 39-41).

Introduction:The introduction primarily cites and discusses mammalian limb studies, despite the relevance of the suggested findings beyond mammals and forelimbs. Furthermore, relevant work exists in other tetrapod clades and structures related to evolutionary rates of later-developing traits and proximo-distal variation.

We followed the recommendations and modified the introduction to discuss tetrapods, instead of focusing solely on mammals. We included references on the limb morphology of lizards (Grizante et al. 2010; *J. Evol. Biol.*; Kohlsdorf et al., 2001, *J. Morphol.*; Rothier et al. 2017 *J. Morphol.*; Rothier et al. 2022 Proc. B), salamanders (Ledbetter and Bonett 2019 *J. Evol. Biol.*) and frogs, particularly highlighting previous findings of the proximo-distal variation of limb segments in anurans (Stepanova and Womack 2021, *Evol*.) (lines 57-58 and 92-95). We also included examples of other structures for which adult variation is predicted by developmental timing including skull bones (Bardua et al. 2021, *Nat. Commun.*; Fabre et al. 2020, *Nat. Ecol. Evol.*) and vertebrae (Adler et al., 2022, *Evol.*) in amphibians (lines 77-81). Finally, we considered it to be relevant to discuss and compare our findings with anurans, highlighted in the discussion at lines 202-215.

Results:When explaining stationary variances you state "which represents the expected variation when the process is in equilibrium and summarizes the relative influence of stochastic factors in evolution". It would be really helpful to explicitly say how this measure summarizes the relative influence of stochastic factors by stating something like "a lower stationary variance value means X whereas a higher stationary variance means Y".

We modified the sentence and removed ‘the influence of stochastic factors’ as it was not notably clear. It reads now as follows (lines 166-172): “Traits evolving under an OU process change at a given step variance (σ^2^) with a strength of constrains (α) towards an adaptive optimum (θ). We interpreted the tempo of evolution of traits considering the mean stationary variance (σ^2^/2α) of each bone, which is a measure of rate under the OU process (Hunt, 2012). The stationary variance represents the expected variation when the OU process is in equilibrium (i.e., around the optimum): the higher the stationary variances, the greater is the phenotypic change around the trait optimum”.

Discussion:Line 235: rephrase "or both in combination therefore".

The whole paragraph was excluded.

Line 327: space needed before citation "species(Weaver and Grossnickle, 2020)"

Addressed.

Finally, although the evolutionary patterns here are interesting there is a missed opportunity with this extensive sampling to test for effects of ecology, locomotion, and/or digit number variation on the patterns observed. This would have provided more clear evidence (or lack of) for some of the suggestions in the discussion.

We strongly agree with the reviewer, and we are already addressing the relationships of limb morphology with ecology and locomotion in two other ongoing studies, using 200 more species that were not included in this work. The present manuscript is the first work of a series that investigate the morphological evolution of limbs in mammals. In the present study we tested the idea that timing of development may be an important driver of variation in the mammalian limb, irrespective of variation in locomotor style or habitat use. The next studies will focus on how limb morphology varies between groups of species with specific ecologies and locomotor styles allowing us to test explicit, a priori hypotheses like we did here for development.

Methods:Although all bones can be relatively scaled, do you think differences among bones in their original size and shape, and thus, differences among bones in the ultimate spacing of the landmarks, could affect the variation in shape seen among structures? Specifically, a smaller, wider bone when scaled equally to a long thin bone might create really different potentials for measured shape change when you think about how those landmarks will vary across different bone shapes in each case. If this influences the analysis/results, it could be a confounding factor since the bones get smaller from proximal to distal.

We acknowledge the concern. However, in comparing the variation between bones, we did not scale all structures to the same size. Each linear measurement obtained was corrected by body size (calculated as the geometric means of body mass plus all the measurements), implying that bone proportions were maintained along the proximo-distal axis. In fact, the traits that vary the most are the metacarpal and the phalanx lengths (variances per measurement illustrated in the graph in Author response image 1), and not the smallest distance values such as heights and widths, showing that variation in relative bone proportion is driving the diversity of the autopod.

**Author response image 1. sa2fig1:** 

Animals having small body sizes – and consequently the smallest bones – were measured digitally, so we were able to zoom to visualize distal bones, avoiding measurement errors due to scale. So landmark spacing is ‘controlled for' by the resolution of the images for smaller specimens. Because we measured each specimen twice, we were able to calculate the standard error of each distance acquired, and we detected that measurement error is similar for all distances. In fact, the landmarks chosen to describe the morphology of the metacarpal and the phalanx are easily identified across species (as described in Appendix 2-table 1 and Figure 2), and they also exhibit high repeatability when taking the standard errors into consideration. In larger animals, measured by caliper, the structures were easy to manipulate and did not show high error measurements after comparing the duplicates. For example, the percentage of the standard error obtained from the duplicates for one of the smallest animals studied, the shrew *Sorex cinereus* (average weight 4.2g) was just of 0.05% for the humerus length (5.67 ± 0.003 mm) and 0.75% for the phalanx length (1.34 ± 0.010 mm). Similar error proportions were also obtained for larger animals: the humpback whale (*Megaptera novaeangliae*, 33 tons average weight) had 1.74% error for the humerus (323.18 ± 5.64 mm) and 0.13% for the phalanx length (186.765 ± 0.245 mm), and the Asian elephant (*Elephas maximus*, 3.3 tons) presented 0.43% error of humerus length (700.78 ± 3.03 mm) and 1.13% of phalanx length (59.26 ± 0.68 mm).

I found it a bit confusing how the size was accounted for in these analyses. The size was the mean mass for the genera from a database? Not from the specimens themselves? Would somehow using the size of the bones themselves provide a better proxy for the body size of the actual animal measured?

We acknowledge that the explanation provided in the first version could have caused confusion, so we modified the section to clarify it (lines 386-390; 398-399 and 404-407). Size was calculated as the geometric means of : (1) the linear-scaled body mass taken from PanTheria (from a database, as pointed by the reviewer, and not from the specimen itself since this information is unfortunately not available for most of the individuals in museum collections), or other sources of literature, and (2) the linear measurements of the specimens themselves. Most specimens were identified at the species-level, so the body mass value used corresponded to the species’ taxonomy. Whenever the classification was available only for the genus but not for the species, we calculated the genus average body mass and used this value to calculate the geometric means. Tips were identified on the phylogeny at the genus level as most of the genera were represented by a single species. Those represented by more than one species had their trait means calculated and used in the downstream analyses.

Reviewer #2 (Recommendations for the authors):Line 1: In the title and main text you use the phrase "evolutionary versatility." I like it, but it's also a little vague, so it'd help if you provide your definition of "evolutionary versatility" somewhere in the text. You have the phrase in parentheses after "stationary variance" in Figure 3E, which implies that it's mostly referring to the stationary variance results, but this isn't clear in the text unless I missed an explanation. Also, you could consider replacing "evolutionary versatility" with the term "evolvability," if you think they're synonymous (e.g., see Love et al. 2021 Paleobiology, and a couple of your references that have "evolvability" in their titles). On a broader note, you often mention "constraints," but you might consider putting greater emphasis on the 'evolutionary versatility (or evolvability) of the distal elements' rather than the 'constraints on proximal elements.' Constraints are often inferred from the lack of something (negative evidence), but it's hard to tell whether it's truly constraints that have led to the lack of some morphologies. See Salazar-Ciudad (2021 Biol. Direct): "The role of development in evolution should be described in an exclusively positive way, as the process determining which directions of morphological variation are possible, instead of negatively, as a process precluding the existence of morphological variation we have no actual reason to expect." However, maybe this is all just semantics, so feel free to ignore this if you'd like.

We agree that “versatility” is a rather vague way to refer to stationary variance. We rephrased the title to “Mammalian forelimb evolution is driven by a gradient of increasing proximal-to-distal morphological diversity”, removing the mention of “evolutionary versatility”. In the main text, we modified it to “lability”. As pointed out by Love et al. (2021), the term “evolvability” carries different connotations depending on the field of study. The term is traditionally used in the quantitative genetics to refer to the ability of a population to evolve towards the direction of selection (see Wagner and Altenberg 1996 *Evol.*; Hansen and Houle 2008 *J. Evol. Biol.*). Because our study is at a macroevolutionary scale, and not at the population level, we preferred not to use the term “evolvability” to avoid misinterpretation of our results.

Lines 35: You may want to explicitly state that "more integrated" refers to more within-element integration. Otherwise, it could be interpreted as between-element integration.

Addressed.

Line 125: You say "within-segment integration," but would "within-element" be more accurate? You report different integration results for the metacarpal and phalanx.

The observation is accurate, we modified the sentence accordingly.

Line 146: Because your Methods are at the end, you may want to mention here what models were outperformed by the OU model. Otherwise, it's a little confusing to have "better" in the sentence without saying what it's better than.

Addressed.

Line 153: It'd be helpful for readers like me who don't have a strong stats background if you better explained "determinants" and "traces". You could do it either here or in your Methods, and/or you could cite a paper or two where they're better explained.

We included an explanation on matrix traces and determinants in the methods (lines 432-436).

Line 179: Tukey's test was the post hoc test after the ANOVA, correct? It may be worth mentioning the ANOVA here.

The reviewer is correct, this has been modified.

Line 198-201: As I note in the public review, your results are likely influenced to some degree by your choice of measurements. You may want to add clarifiers to sentences like this, such as "Based on linear measurements of limb elements, we show a general.…". Or something along those lines.

We changed the phrase following the recommendation.

Line 201: Yes, but your results are also consistent with the hypothesis that the autopod is more functionally specialized and therefore has a greater diversity of forms (and greater integration). This is an example of a spot where I recommend first discussing functional diversity because I think it offers a better explanation for your results.

We modified the paragraph, following the major recommendations (but see response to comment #11).

Line 229: "as whales and bats" implies that if marsupials evolved to be aquatic or power-fliers then they'd be whales and bats. But those are placental groups. Maybe you mean something like "comparable to whales and bats in Placentalia".

This paragraph was excluded from the manuscript.

Line 238: Pevsner et al. 2022 are cited here, but I don't see it in the References section.

Addressed.

Line 302-306: I really like this summary sentence – you lead with function, and then mention development, which is more in line with my interpretation of results (see comments above). I recommend moving statements like this to the forefront of the paper.

We addressed the recommendations accordingly.

Line 384: Maybe "most part" should just be "most"?

Addressed.

Line 389: I like that you included body mass, but one issue is that it's from a different source (PanTheria) rather than being the weights of the actual specimens you measured, and this introduces a source of error. The masses provided in PanTheria are likely very different than the masses of the individuals you measured. If, for example, PanTheria says a species is 300g, but you measured a smaller individual that's only 100g, then this could bias all of the size-corrected measurements for that species. This is likely to have the biggest influence on smaller taxa/individuals, which can vary considerably in size even throughout a day (e.g. insectivorous bats can double in weight after a night of feeding). I doubt that this has a significant impact on your results, but I recommend justifying the use of body mass in your size-correction calculations. Also, you could go a step further and re-analyze your data without including body mass during size correction to see what influence it has on overall results.

We acknowledge the concern. Indeed, we could not recover the individual body masses for the majority of species included as this information is not available for most museum specimens. This is also why geometric means were calculated to conduct the size correction. The geometric means include values from the individuals themselves and the average species body mass (only one out of 20 variables). We included this justification in lines 404-407. As suggested, we also repeated the empirical analyses without using the body mass during size correction. As expected by the reviewer, it did not impact on the results obtained (this information now included in lines 417-421 and in the Supplementary file 2c and 2d):

Lines 391 and 401: Be careful when using "size" or "mass". They're not quite synonymous. On Line 391 you say "body size," but I think you mean mass. And your PGLS regressions are for correcting for size, so on Line 401 I'd say "body size PGLS" rather than "body mass PGLS".

We acknowledge the observation and modified the sentences accordingly.

Line 395-402: I got a little confused here. It's not immediately clear to me why you needed to simulate data. Couldn't the downstream analyses be conducted with the original dataset (or residuals after the regression) and then error or significance be calculated with the simulated datasets? Or are the simulations for ancestral state reconstructions (like SIMMAPS)? A little more information would be helpful.

This was indeed not clearly stated in the original manuscript. The analyses were conducted according to the first interpretation provided by the reviewer. It is possible to calculate the values of trace, determinant, integration, and stationary variances without simulating the datasets, but the simulations are essential to compare the significance. We did estimate the empirical values, and then we simulated the 100 datasets incorporating the parameter estimates from the best fit model on the original data (i.e., an OU process). We did not include the empirical results in the last manuscript version because we showed the distributions of the simulated data instead, but we recognize the importance to include these values for comparison. Therefore, in the distribution represented at Figure 3 (i.e., the 100 simulated results for each bone), we additionally included reference lines indicating these empirical values. These values are indicated for the trace, determinant, and integration, and they are now also available in the SI (Appendix 2 – table 3). For the stationary variances, however, the reference line indicates the median of the regressions, and not the empirical value from the MCC tree, using 100 random trees from Upham et al. (2019). We adapted the text for clarification (see lines 143, 150-152 and 421-428).

Line 409: Is there a reason why you used ANOVA instead of phylogenetic ANOVA?

Unlike the previous analyses, it was not possible to compare here the values of interest between taxa, but between bones. Each one of the regressions using the simulated data generated one evolutionary matrix of variance and covariance (all having the same species) – in other words, we had a total of 100 evolutionary matrices describing the variance and covariance of the humerus traits, and another 100 each for the radius, metacarpal and phalanx. From each one of these matrices, we obtained a single value of determinant, trace, and integration. We then used the conventional ANOVAs to estimate the differences of those parameters between the bones. Although it does not consist of a phylogenetic ANOVA, the phylogenetic component from previous matrix estimate is accounted for during the generation of the parameters.

Figure 2: You could color-code the bones to match the colors in Figure 3.

Bones are now color-coded in accordance to Figure 3.

Figure 3: Is there any way to add another part to this figure that illustrates the typical timing of condensation for different bones? It could be fairly conceptual, just showing the relative timings. I think that would help you to better emphasize/visualize your conclusions.

We addressed the recommendation and added a scheme representing the typical timing of limb bone condensation in Figure 2, alongside the illustration of the measurements obtained. We also indicated the elements we analyzed in the scheme, using the same colors used in Figure 3.

Figure 3D: I recommend adding a unit for integration (maybe "values of eigenvalue dispersion", Line 165) in parentheses after "Integration" in the figure and/or in the caption.

Addressed.

Reviewer #3 (Recommendations for the authors):Given that the paper will ultimately not discuss the hindlimb at all, I am not sure the section in the introduction from lines 98 to 114 is entirely relevant. Certainly discussion of fore and limb serial homology is not relevant to the introduction of this paper.

The comparison between fore and hindlimbs aims to introduce the importance of functional and developmental factors (attributed to the serial homology of structures) on the variation of limb integration. We included “For example” at the beginning of this section to clarify the explanatory tone of the sentence. We consider the section is relevant because it provides reasoning for elements later presented in our discussion, so we decided to keep it in the introduction.

The authors mention they took multiple (two) sets of measurements of each bone to account for measurement error. However, the authors included measurements from three different sorts of data (physical specimens, microCT scans, and surface scans) over a huge size range (8 orders of magnitude from 3 g to 30 000 000 g based on the supplemental data). As such the possibility of size-related and measurement method related both relative and absolute measurement error needs to be considered. Do the authors have any numerical data, or data from published studies, addressing the issue of measurement error in large taxonomic samples including a mix of physical and digital specimens?

As previously mentioned (see answer to comment #8 from Reviewer 1), the error estimate (that is, the percentage describing the standard error according to the trait mean) is very low and similar across all species, rarely exceeding 1.5% irrespective of the animal’s size and the measurement method. Error estimates of all distances were checked for each specimen prior to comparing the means between groups. We have generated one file for each specimen (that is 670 files) detailing the distances obtained for the first and the second acquisition, as well as their corresponding standard error, which can be made available upon request to the first author (PSR). We included more detailed information about measurement error estimate in lines 384-386.

Finally, the concern involving the error between the methods of data acquisition is quite relevant. Previous studies have compared the average percentage differences from skeletal measurements of a same individual using different methods, including the full body CT scan with soft tissues, the CT scan of the dry skeletal elements, and caliper inference in the dry bones (see Stull et al. 2014 Forensic Science International; Robinson and Terhune 2017 American Journal of Biological Anthropology; Ismail et al. 2019 Journal of Forensic and Legal Medicine). Average percent differences are always minimal across the data sources, showing that all the methods are reliable to provide repeatable linear distances.

There are 638 species represented in the dataset, yet 670 specimens. Thus almost all species are represented by a single specimen. Indeed only 33 species in the dataset are represented by multiple specimens, and that is never more than 2. Furthermore, it appears for much of the analysis the unit of analysis is the genus mean. Thus what is the rationale for those duplicate measurements in 33 cases?

While structuring the data acquisition, our goal was to capture at least 50% of the genus diversity per family, aiming to provide a broad quantification of the morphological variation across the mammalian tree of life. However, species diversity is not evenly distributed across families, with many groups having multiple species and other ones exhibiting very few taxa. The rationale for using multiple specimens to represent a single species precisely considered the species diversity per family, in addition to the availability of individuals in the collections and at online repositories. For example, the only living species representing the family Ornithorhynchidae is the platypus *Ornithorhynchus anatinus*, and we have measured two individuals of this species to improve the morphological characterization for this taxon. We considered that in groups we were able to sample many species, the taxon diversity was satisfactorily described, and we did not include additional individuals because of the time required to provide the measurements (specifically considering that all morphological data was gathered by a single researcher). The same reasoning was applied to other groups with few species, as Procaviidae (3 genera), Orycteropodidae (1 species), Dugongidae (one living species), etc. However, for some other families also represented by few species (for example Elephantidae, Tarsipedidae, etc.), we were not able to find additional specimens with the required skeletal elements to be included in our dataset. We also randomly added a second specimen to some few species belonging to diverse groups (*Mus musculus*, Muridae and *Glossophaga soricina*, Phyllostomidae) because of their abundance in online repositories thus allowing to assess the within-species variation more broadly.